# From Extrinsic to Intrinsic: Geodesic-Guided Representation Learning for 3D Geometric Data

**Yuming Zhao** [1]  **Junhui Hou** [1]  **Qijian Zhang** [2]  **Jia Qin** [3]  **Ying He** [4]

## Abstract

Geometric analysis fundamentally distinguishes between *extrinsic* and *intrinsic* perspectives. The dominant paradigm in current 3D representation learning relies on either extrinsic spatial structures or high-level semantics, struggling to capture the essence of shape identity and underlying manifold topology. To bridge this gap, we introduce a novel 3D representation learning paradigm, namely **PRISM**, for **P**re-training, which learns isometric embeddings by **R**ecovering the **I**ntrinsic **S**urface geodesic **M**etric. PRISM incorporates a topology-enforcing objective that explicitly constrains the structure of latent space, alongside a specialized two-stage training recipe mitigating sample imbalance inherent in the distribution of geodesic distances. Experiments demonstrate that our approach shows satisfactory accuracy, robustness, and high efficiency in geodesic distance prediction and achieves superior performance across diverse downstream tasks, including shape recognition, surface parameterization, and non-rigid correspondence. The code will be publicly available at https://github.com/AidenZhao/PRISM.

## 1. Introduction

Self-supervised representation learning has emerged as a pivotal research direction in processing and understanding 3D geometric data. Developing powerful backbone feature extractors through pre-training is a critical prerequisite for building scalable foundation models and achieving superior generalization across diverse downstream applications.

In recent years, a rich variety of 3D pre-training frameworks have been investigated to continuously push the boundary of representation quality and transferability, built upon contrastive learning (Xie et al., 2020), masked modeling (Wang et al., 2021; Pang et al., 2022), cross-domain interaction (Afham et al., 2022; Zhang & Hou, 2024), etc. Generally, existing pre-training objectives are driven by either extrinsic 3D spatial structures and/or high-level semantic information, producing superior performance on downstream tasks such as classification and segmentation. However, due to the lack of modeling and learning of intrinsic geometric properties, existing frameworks still struggle to capture the essence of shape identity, understand the underlying manifold topology, and produce high-quality fine-grained geometric features. Consequently, these approaches typically suffer from suboptimal performance when evaluated on geometry-sensitive tasks that are grounded in intrinsic manifold structures or require robustness against deformations and pose variations, such as surface parameterization and non-rigid shape correspondence.

As a fundamental Riemannian metric, **geodesic distance** serves as the definitive intrinsic characterization of geometry on 3D curved surfaces, remaining invariant under isometric deformations. Accurately and efficiently computing geodesic distances is a central and long-standing problem in the field of computational geometry. While conventional geometry processing techniques deliver high precision and provide theoretical guarantees, they typically suffer from prohibitive computational bottlenecks. More recently, learning-based approaches have emerged as a promising alternative, offering substantially faster query speeds as well as greater flexibility. Despite the remarkable advances, existing approaches are still faced with obvious limitations in terms of robustness (GeGNN (Pang et al., 2023)) and cumbersome pre-computation (NeuroGF (Zhang et al., 2023)).

Drawing inspiration from *Nash Embedding Theorem* (Nash, 1956) (*i.e., any Riemannian manifold can be isometrically embedded into a higher-dimensional Euclidean space while preserving geodesic distances*), we introduce geodesic distance prediction as a core pretext task for 3D geometric data pre-training. Our objective is to learn a latent feature space that approximates isometry with respect to the original intrinsic geometry, such that the backbone feature extractor

[1]Department of Computer Science, City University of Hong Kong, Hong Kong, China [2]Bambu Lab, Shenzhen, China [3]Meshy AI, California, USA [4]College of Computing and Data Science, Nanyang Technological University, Singapore. Correspondence to: Junhui Hou <jh.hou@cityu.edu.hk>.

*Proceedings of the $43^{rd}$ International Conference on Machine Learning*, Seoul, South Korea. PMLR 306, 2026. Copyright 2026 by the author(s).

is enforced to capture both intrinsic manifold structure and fine-grained geometric details.

Technically, we employ a powerful point transformer (Wu et al., 2024) architecture with high efficiency and scalability as the target backbone. For geodesic-guided representation learning, we resort to a two-stage workflow that combines a pre-training stage constrained with geodesic structure consistency and a fine-tuning stage introducing importance sampling to mitigate sample imbalance. The pre-trained model achieves high-precision geodesic distance prediction with only 3D point clouds as input.

Empirically, we evaluate the effectiveness of our proposed geodesic-guided 3D pre-training framework by performing task-specific fine-tuning. For high-level semantic-oriented tasks, we conduct shape classification and part segmentation. For fine geometric details-centric tasks, we conduct fixed-boundary surface parameterization and non-rigid shape correspondence. Extensive experiments demonstrate that our approach consistently delivers competitive performance.

In essence, the main contributions of this work can be summarized as follows.

- We propose a novel 3D representation learning paradigm guided by intrinsic geodesic properties for learning robust and fine-grained geometric features.

- Our framework serves as a high-precision geodesic distance predictor with unstructured point clouds as input.

- Particularly, thanks to intrinsic geometry pre-training, we achieve the **first** successful training of a feed-forward surface parameterization model.

## 2. Related Work

### 2.1. Representation Learning of 3D Geometry

**Contrastive Learning.** The essential goal of contrastive learning approaches is to measure the feature-level similarity and dissimilarity between positive and negative samples. PointContrast (Xie et al., 2020) constructs two point clouds from different perspectives and performs pre-training by measuring the feature similarity between corresponding points. Subsequent studies further investigate more aggressive data augmentation strategies (Wu et al., 2023; Zhang et al., 2021), self-distillation (Wu et al., 2025), cross-modal interaction (Afham et al., 2022; Zhang & Hou, 2024; Zhang et al., 2025a), etc.

**Masked Modeling.** Inspired by the success of masked 2D image autoencoders (He et al., 2022), a variety of adaptations are explored for 3D data pre-training. The representative works of Point-BERT (Yu et al., 2022) and Point-MAE (Pang et al., 2022) perform masked reconstruction on point clouds. TAP (Wang et al., 2023) and Ponder (Huang

et al., 2023) drive pre-training by generating 2D projections of 3D point clouds. Point-M2AE (Zhang et al., 2022) introduces a hierarchical architecture progressively modeling geometric and feature information from local to global scales. Joint-MAE (Guo et al., 2023) designs joint encoders and decoders between 2D image and 3D point cloud modalities.

In addition, there also emerge more novel self-supervision paradigms such as adopting denoising mechanisms (Zheng et al., 2024; Zhang et al., 2025b; Chen et al., 2025).

### 2.2. Geodesics Computation

Computing geodesic distances and paths on 3D surfaces has been extensively studied for several decades (Mitchell et al., 1987), resulting in a large body of literature (Bose et al., 2011; Crane et al., 2020). Existing methods can be broadly categorized into two categories: traditional methods and learning-based methods.

**Traditional Methods.** Classical exact algorithms compute polyhedral geodesics by propagating continuous wavefronts, commonly known as continuous Dijkstra methods (Mitchell et al., 1987; Chen & Han, 1990; Xin et al., 2012; Surazhsky et al., 2005; Ying et al., 2014; Xu et al., 2015; Qin et al., 2016; Sharp & Crane, 2020). These methods are generally robust and capable of handling low-quality meshes, but they are computationally expensive. In applications that require frequent point-to-point queries, preprocessing techniques such as geodesic graphs or Steiner-point constructions (Ying et al., 2013; Adikusuma et al., 2020; Meng et al., 2022) are commonly used to spread out the computational cost. PDE-based techniques compute distance fields by solving or approximating the Eikonal equation. Representative methods include fast marching on triangulated domains (Kimmel & Sethian, 1998; Sethian & Vladimirsky, 2000) and heat-flow-based approaches (Crane et al., 2013; Sharp et al., 2019; Tao et al., 2021). PDE methods typically provide only first-order approximations of geodesic distances and often rely on high-resolution and high-quality input data to achieve satisfactory accuracy.

**Learning-based Methods.** More recently, learning-based geodesic computation approaches are increasingly revealing new potential. NeuroGF (Zhang et al., 2023) introduces a deep implicit neural geodesic field that encodes the geodesic structure of a given 3D shape, enabling fast inference-time queries of arbitrary point-to-point geodesic distances and shortest paths, together with extensions to generalizable frameworks with learned 3D shape encoders. GeGNN (Pang et al., 2023) employs a graph neural network that maps mesh vertices into the high-dimensional embedding space, and answers distance queries by applying a lightweight decoder to the embeddings of queried points. More recently, LiteGE (Adikusuma et al., 2026) challenges the need for costly shape encoders. Instead, it canonicalizes

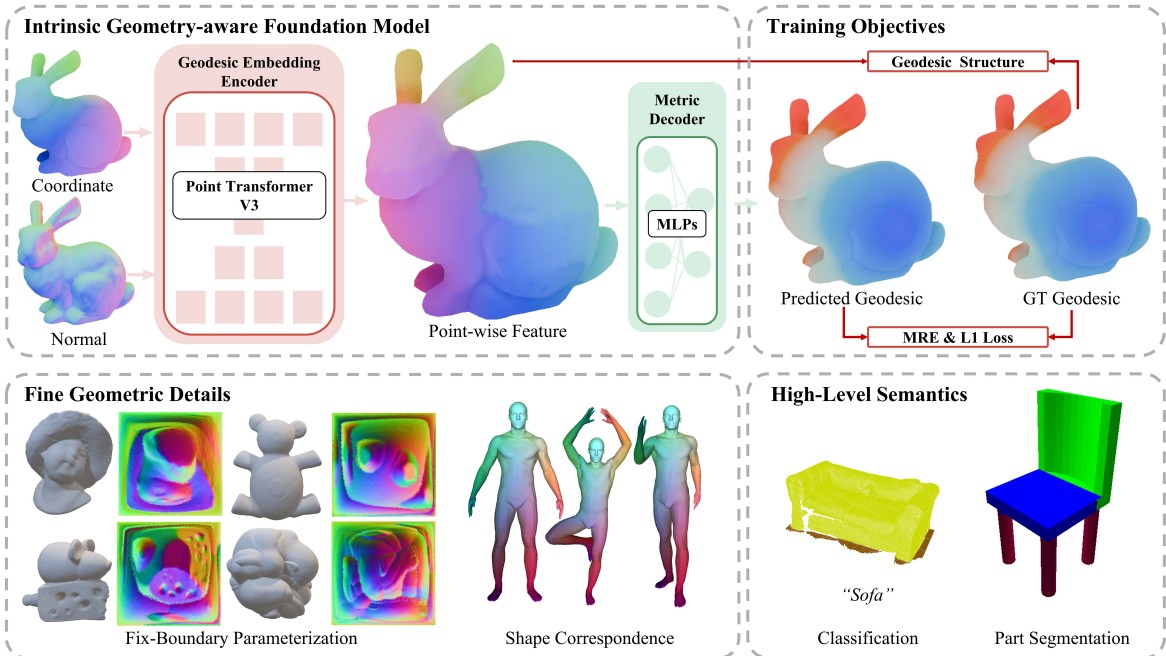

**Figure 1.** The overview of **PRISM**, including an intrinsic geometry-aware foundation model and a geodesic-driven training objective composed of geodesic structure and prediction. Our PRISM effectively facilitates downstream tasks that focus on fine geometric details and high-level semantics.

input shapes and builds compact, category-aware descriptors by applying PCA to unsigned distance field samples at informative voxels and then predicts geodesic distances using a small inference network.

## 3. Proposed Method

We introduce a novel 3D representation learning paradigm, namely **PRISM**, for **P**re-training, which learns isometric embeddings by **R**ecovering the **I**ntrinsic **S**urface geodesic **M**etric. An overview of PRISM, is shown in Figure 1.

### 3.1. Motivation

**Geodesic Embedding.** The fundamental goal of representation learning on 3D surfaces is to find a mapping that preserves geometric information. The *Nash Embedding Theorem* (Nash, 1956) provides a rigorous theoretical foundation for this pursuit. It states that every Riemannian manifold can be isometrically embedded into a Euclidean space of sufficiently high dimension.

Inspired by this theorem, we seek to learn a non-linear mapping $\Phi : \mathcal{M} \to \mathbb{R}^k$ that embeds the input manifold $\mathcal{M}$ (represented by a 3D point cloud) into a high-dimensional feature space $\mathbb{R}^k$. Our objective is to ensure that this embedding is **approximately isometric**, meaning the metric structure of the feature space reflects the intrinsic metric of the input surface.

**Geodesic Distance as the Intrinsic Metric.** A 3D point cloud $\mathcal{P}$ is often treated as a set of coordinates in $\mathbb{R}^3$. However, these coordinates usually describe the extrinsic geometry. The essence of the shape lies in its intrinsic geometry properties that depend only on the surface itself, not on how it is folded or bent in space.

The geodesic distance $d_G(\cdot, \cdot)$ is the natural realization of the Riemannian metric on the manifold. Unlike the classic Euclidean distance $d_E(\cdot, \cdot)$ which cuts through the ambient space, $d_G$ measures the shortest path along the surface. By training our model to predict geodesic distances, we explicitly force the learned feature space to respect the underlying manifold structure. This serves as a robust foundation for fine-grained geometric processing tasks, where understanding topological connectivity is crucial.

### 3.2. Network Architecture

We employ Point Transformer V3 (PTv3) (Wu et al., 2024) as our feature representation backbone. PTv3 is chosen for its efficiency in processing large-scale point clouds and its ability to capture both local geometric details and global context through self-attention mechanisms. Given an input point cloud $\mathcal{P} \in \mathbb{R}^{N \times 3}$ of $N$ points, the backbone maps each point $p_i$ to a high-dimensional feature vector $\mathbf{f}_i \in \mathbb{R}^k$.

$$\mathbf{F} = \text{PTv3}(\mathcal{P}), \quad \text{where } \mathbf{F} = \{\mathbf{f}_1, \dots, \mathbf{f}_N\}. \quad (1)$$

**Geodesic Prediction Head.** We designed a metric decoder to represent the metric between two points on $\mathbb{R}^k$, constraining it to be as close as possible to the geodesic distance between the two points on $\mathcal{M}$. To predict the intrinsic dis-

tance between any pair of points $(\mathbf{p}_i, \mathbf{p}_j)$, we employ a symmetric predictor that operates on the feature difference. Specifically, we compute the absolute difference between their feature vectors:

$$\mathbf{h}_{ij} = |\mathbf{f}_i - \mathbf{f}_j| \in \mathbb{R}^k. \tag{2}$$

This difference vector is then fed into a 3-layer Multi-Layer Perceptron (MLP) that maps the feature difference to a scalar value $R_{ij}$, representing the estimated geodesic distance, formulated as:

$$\hat{d}_{ij} = \text{MLP}(\mathbf{h}_{ij}). \tag{3}$$

Using the absolute difference ensures the prediction is symmetric, i.e., $\hat{d}_{ij} = \hat{d}_{ji}$, respecting the symmetry of the distance metric.

### 3.3. Pre-training Objective

**Geodesic Regression Loss ($\mathcal{L}_{L1}$).** First, we apply a standard L1 loss to directly minimize the absolute error between the predicted and the ground truth geodesic distances:

$$\mathcal{L}_{L1} = \frac{1}{|\mathcal{S}|} \sum_{(i,j) \in \mathcal{S}} |\hat{d}_{ij} - d_G(p_i, p_j)|, \tag{4}$$

which ensures that the model learns the correct scale of the manifold.

**Mean Relative Error Loss ($\mathcal{L}_{MRE}$).** Since geodesic distances can vary significantly in magnitude (from local neighborhoods to global extrema), an L1 loss may be dominated by large distances. To ensure the model captures fine-grained local geometry as well as global structure, we incorporate a relative error term:

$$\mathcal{L}_{MRE} = \frac{1}{|\mathcal{S}|} \sum_{(i,j) \in \mathcal{S}} \frac{|\hat{d}_{ij} - d_G(p_i, p_j)|}{d_G(p_i, p_j) + \epsilon}, \tag{5}$$

where $\epsilon$ is a small constant for numerical stability.

**Geodesic Structure Consistency Loss ($\mathcal{L}_{struct}$).** Beyond regressing exact values, we explicitly constrain the structure of the latent space to reflect the relative order of distances on the manifold. We employ a continuous ordinal loss to align the pairwise feature distances with the geodesic distances. Specifically, for two pairs of points $(i,j)$ and $(u,v)$, we define the geodesic difference $\Delta d_G = d_G(p_i, p_j) - d_G(p_u, p_v)$ and the feature distance difference $\Delta d_\Phi = \|\mathbf{f}_i - \mathbf{f}_j\| - \|\mathbf{f}_u - \mathbf{f}_v\|$.

This loss enforces that the sign of the feature difference matches the sign of the corresponding geodesic difference. To ensure differentiability, we approximate the sign function using a scaled hyperbolic tangent ($\tanh$). The loss is formulated as the L1 distance between the true sign of geodesic differences and the soft sign of feature differences:

$$\mathcal{L}_{struct} = \frac{1}{|\mathcal{Q}|} \sum_{((i,j),(u,v)) \in \mathcal{Q}} |\text{sgn}(\Delta d_G) - \tanh(\alpha \cdot \Delta d_\Phi)|, \tag{6}$$

where $\alpha$ is a scaling factor that controls the steepness of the approximation, acting as a soft-sign function. This ensures that if pair $(i,j)$ is geodesically closer than pair $(u,v)$, their representations in feature space will also be closer, preserving the ordinal structure of the manifold.

### 3.4. Training Strategy

To enhance convergence speed and address the imbalance in geodesic distance distributions, we propose a two-stage workflow: (**1**) geodesic structure warm-up and (**2**) importance sampling fine-tuning.

**Geodesic Structure Warm-Up.** In the initial training phase, the primary goal is to rapidly guide the model towards a topologically meaningful feature space. To this end, we incorporate the geodesic structure consistency loss ($\mathcal{L}_{struct}$) with a time-decaying weight. By placing a heavy emphasis on structural consistency early on, we enforce a global ordering constraint that prevents the model from collapsing into local minima driven solely by distance regression. We define a dynamic weight $\lambda_3(t)$ for $\mathcal{L}_{struct}$ that decreases over epochs:

$$\lambda_3(t) = \lambda_{init} \cdot \left(1 - \frac{t}{T_{warmup}}\right), \tag{7}$$

where $t$ is the current epoch and $T_{warmup}$ is the duration of the warm-up phase. This strategy accelerates convergence by first establishing the correct "shape" of the latent manifold before refining the precise metric values.

In all, the overall training objective can be formulated as:

$$\mathcal{L} = \lambda_1 \cdot \mathcal{L}_{L1} + \lambda_2 \cdot \mathcal{L}_{MRE} + \lambda_3(t) \cdot \mathcal{L}_{struct}. \tag{8}$$

**Importance Sampling Fine-Tuning.** As illustrated in Fig. 2, the distribution of geodesic distances between paired points in 3D shapes typically follows a normal-like distribution, where mid-range distances are abundant, but short-range and long-range distances are rare. Hence, standard uniform sampling causes the model to underfit these distant pairs, leading to inaccurate global feature representations. To mitigate this, we introduce an Importance Sampling Fine-Tuning phase. We pre-compute the empirical probability density function $P(d)$ of the geodesic distances in the training set. During fine-tuning, we sample point pairs $(i,j)$ with probability inversely proportional to their occurrence:

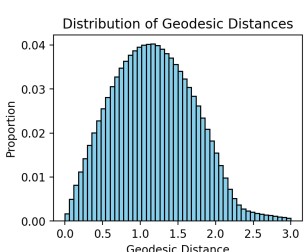

*Figure 2.* The distribution of geodesic distance values.

$$w_{sample} \propto \frac{1}{P(d_G(p_i, p_j))}. \tag{9}$$

*Table 1.* Quantitative comparison of geodesic distances estimated by different methods. For all metrics, the smaller, the better. The best results are highlighted in **bold**. Note that [S/M/L] NeuroGF is **per-scene overfitting**, where for each 3D shape, a portion of ground-truth geodesic distances have to be pre-computed to train the network.

| Method | Input | MRE (%) | L1 (%) | Time (s) |
|---|---|---|---|---|
| **Traditional Methods** | | | | |
| HM | Mesh | 2.52 | 2.71 | 79.7 |
| DGG | Mesh | **0.25** | **0.27** | 37.8 |
| EEM | Mesh | 8.73 | 8.41 | 45.7 |
| FPGDC | Mesh | 2.49 | 2.31 | 12.1 |
| **Learning-based Methods** | | | | |
| LiteGE | 10.81 | 3.91 | **0.3** | |
| [O] GeGNN | Mesh | 23.2 | 20.8 | 1.4 |
| [R] GeGNN | Mesh | 4.43 | 3.26 | 18.9 |
| [S] NeuroGF | Points | 3.12 | 2.93 | 60 |
| [M] NeuroGF | Points | 1.84 | 1.69 | 120 |
| [L] NeuroGF | Points | 0.52 | 0.48 | 600 |
| **Ours** | Points | 3.87 | 2.75 | 0.5 |

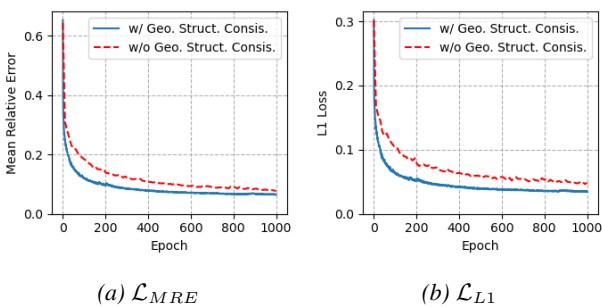

*(a) $\mathcal{L}_{MRE}$*         *(b) $\mathcal{L}_{L1}$*

*Figure 3.* Ablation of Geodesic Structure Consistency results on $\mathcal{L}_{MRE}$ and $\mathcal{L}_{L1}$.

*Table 2.* Ablation of importance sampling fine-tuning results on $\mathcal{L}_{MRE}$ of different geodesic distance in $(\mathbf{0}, \mathbf{1}]$, $(\mathbf{1}, \mathbf{2}]$ and $(\mathbf{2}, \mathbf{3}]$ .

| Setting | $(\mathbf{0}, \mathbf{1}]$ | $(\mathbf{1}, \mathbf{2}]$ | $(\mathbf{2}, \mathbf{3}]$ |
|---|---|---|---|
| w/o Fine-tuning | 5.1% | 1.8% | 2.9% |
| w/ Fine-tuning | 3.9% | 1.7% | 2.4% |

In this phase, we disable the structure loss (setting $\lambda_3 = 0$) and focus exclusively on the regression objectives ($\mathcal{L}_{L1}$ and $\mathcal{L}_{MRE}$). By prioritizing rare, long-range distances, we fine-tune the model to achieve high precision across the entire spectrum of the manifold.

# 4. Experiments

## 4.1. Implementation Details

We collected a refined subset of the classic ShapeNet (Chang et al., 2015) repository in the self-supervised pre-training phase, consisting of 15,000 training samples and 2,000 testing samples. To ensure high-quality geometric supervision, all the original mesh models were pre-processed through watertight manifold reconstruction and uniform remeshing, then normalized into a unit sphere. We applied the classic MMP (Mitchell et al., 1987) algorithm to produce ground-truth geodesics. For each training shape, we randomly sampled 500 source points to deduce geodesic distance fields.

The overall pre-training process is composed of two stages. In the first 1,000 epochs, we performed the Geodesic Structure Warm-Up. Then we performed fine-tuning with Importance Sampling for the subsequent 200 epochs. We adopted the AdamW optimizer with an initial learning rate of 1e-4, which decays to 1e-6 following a Cosine Annealing schedule. Our model is trained on 8 NVIDIA H100 GPUs with the total batch size of 64.

## 4.2. Geodesic Distance Prediction

To evaluate our performance of geodesic distance prediction, we made comparisons with traditional computational approaches (HM (Crane et al., 2013), DGG (Adikusuma et al., 2020), EEM (Panozzo et al., 2013), FPGDC (Shamai et al., 2018)) and state-of-the-art learning-based frameworks (GeGNN (Pang et al., 2023), NeuroGF (Zhang et al., 2023)). We quantitatively measured the metrics of Mean Relative Error (MRE), L1 Error, and average time cost (2000 queries per testing shape model). Considering the specific characteristics of the two learning-based competitors, we adopted different evaluation protocols. Given the sensitivity of GNNs to mesh density, we evaluated GeGNN on both our original testing data ([O] GeGNN) and a separate version remeshed to match the density used in its paper ([R] GeGNN). As NeuroGF relies on per-scene overfitting, the duration of the training stage significantly impacts performance. Therefore, we configured three training durations for comparison: short (*1 minute*, [S] NeuroGF), medium (*2 minutes*, [M] NeuroGF), and long (*10 minutes*, [L] NeuroGF). We reported quantitative comparison results in Table 1.

Traditional computation- and optimization-based methods can usually achieve the highest accuracy and demonstrate the most stable and reliable performance across various scenarios, but they are slower, limited by pre-computation time and relatively slow geodesic query speed. Among learning-based methods, GeGNN is quite sensitive to mesh density due to its GNN architecture. On our dataset, GeGNN almost completely fails, with MRE reaching 23.2%. However, when we remesh the test data to the same resolution as the training data, GeGNN's performance can recover to a normal level. In contrast, NeuroGF, as a neural field method that performs per-scene overfitting for individual models, requires longer training time, but its prediction accuracy can gradually approach extremely high levels comparable to traditional methods. Compared with other methods, our

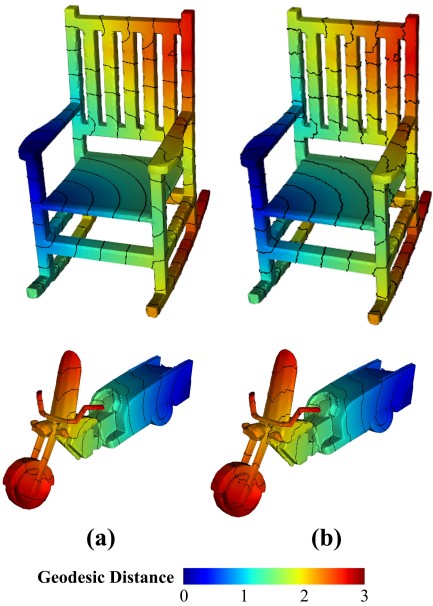

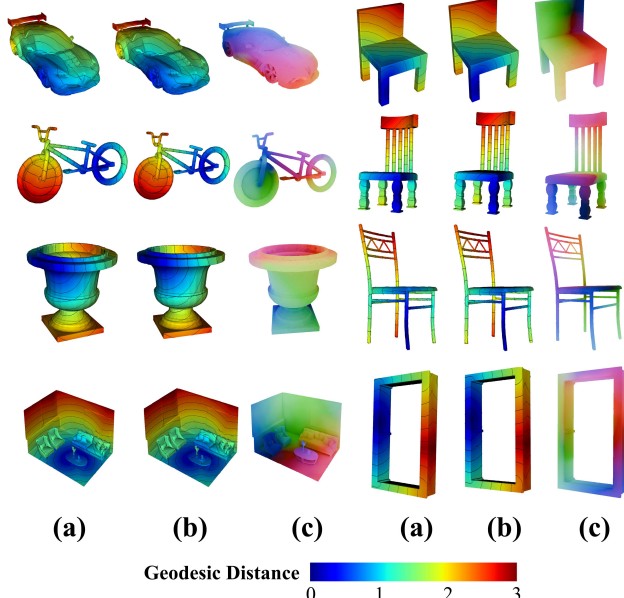

*Figure 4.* Visual results on ablation of importance sampling fine-tuning. (a) w/ importance sampling fine-tuning, (b) w/o importance sampling fine-tuning.

*Table 3.* Comparison of different feature dimensions of our backbone on geodesic distance estimation.

*Figure 5.* Visualization of geodesic prediction results by our methods. (a) Ground Truth by MMP (Mitchell et al., 1987), (b) Geodesic prediction by our method, (c) Point-wise feature visualization by t-SNE. It can be observed that the point-wise features exhibit a structure aligned with the geodesic distance, as demonstrated by the t-SNE dimensionality reduction visualization.

| Setting | # Params. | $k$ | GFLOPs | MRE (%) | L1 (%) |
|---------|-----------|-----|--------|---------|--------|
| Small | 31M | 256 | 82.87 | 6.5 | 5.2 |
| Base | 124M | 512 | 328.95 | 4.4 | 3.2 |
| Large | 494M | 768 | 1295.81 | 3.9 | 2.8 |

approach achieves high accuracy and efficient geodesic distance prediction. Fig. 5 shows a comparison between the geodesic lines predicted by our method and the ground truth, as well as the visualization of the obtained features after t-SNE dimensionality reduction. Our method was trained only on object shapes from ShapeNet, yet it still exhibits a certain degree of generalization capability to inputs of scene type.

In Figure 7, we present a set of results featuring non-watertight and noisy inputs. As illustrated, our method demonstrates strong robustness against both topological defects and sensor noise. In Figure 8, we show a heatmap of geodesic distance and feature Euclidean distance between 50 points on Bunny. It can be seen that the distribution and scale ordinary between geodesics and features are similar.

## 4.3. Ablation Study

To evaluate the necessity and effectiveness of our core designs, we conducted ablation studies on three components: geodesic structure consistency, importance sampling fine-tuning, and different feature dimensions.

**Geodesic Structure Consistency.** We targetedly observed the training curves of MRE and L1 error under the same pre-training setup with and without geodesic structure warm-up. As shown in Figure 3, its integration effectively accelerates the decrease of both MRE and L1 error.

**Importance Sampling Fine-Tuning.** We compared the distribution curves of MRE across different geodesic distance lengths before and after applying importance sampling fine-tuning. As shown in Table 2 and Fig. 4, the accuracy for both short and long geodesic distances can greatly improve after such fine-tuning, effectively mitigating the prediction bias caused by the natural imbalance in the distance distribution.

**Feature Dimension.** Regarding scaling, we evaluated three models of different feature dimensions. As shown in Table 3, increasing the model capacity steadily improves geodesic distance prediction accuracy.

**Geodesic Prediction Head.** In Figure 9, we present a comparative analysis of training dynamics. By comparing our method with a variant that calculates feature-wise Euclidean distance without an MLP, it becomes evident that the latter suffers from convergence issues, whereas our full model converges smoothly.

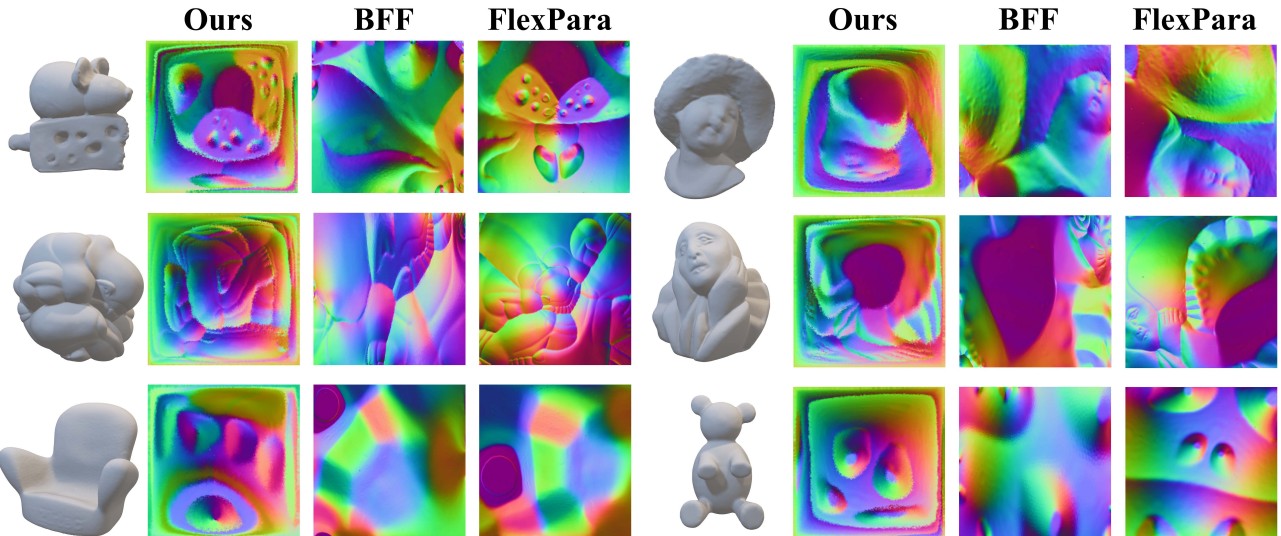

*Figure 6.* Visual comparison of fixed-boundary parameterization results by different methods. From left to right: Ours, BFF (Sawhney & Crane, 2017), and Flexpara (Zhao et al., 2025).

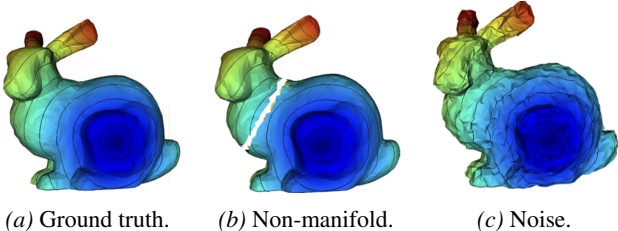

*(a)* Ground truth.  *(b)* Non-manifold.  *(c)* Noise.

*Figure 7.* Geodesic prediction on different input.

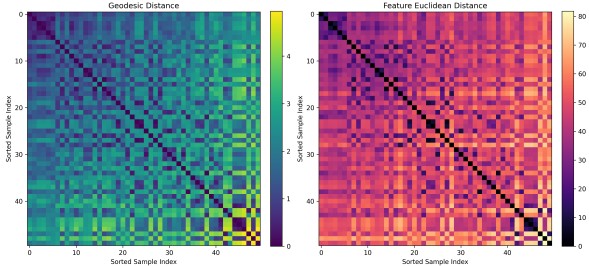

*Figure 8.* Geodesic and feature euclidean distance heatmap.

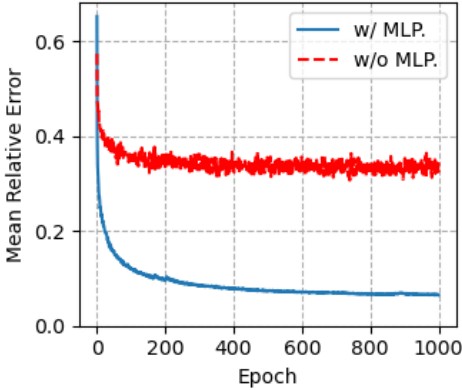

*Figure 9.* Training loss curve on w/ MLP and w/o MLP.

*Table 4.* Quantitative comparison of fixed-boundary surface parameterization. The best results are highlighted in **bold**. Note that FlexPara is **per-scene overfitting** method, which needs a **long time** to train. "M+S": Mesh + Seams.

| Method | Input | Error | Time |
|---|---|---|---|
| BFF (Sawhney & Crane, 2017) | M+S | 7.27 | 4.3 |
| FlexPara (Zhao et al., 2025) | Points | **4.45** | 423 |
| **Ours** | Points | 5.34 | **0.5** |

### 4.4. Downstream Tasks

Generally, we focus on two categories of downstream task scenarios respectively dominated by fine geometric details and high-level semantics. For the former, we evaluated the novel fixed-boundary surface parameterization task and the challenging non-rigid shape correspondence task. For the latter, we evaluated the standard tasks of shape classification and object part segmentation.

**Fixed-Boundary Surface Parameterization.** Parameterization refers to mapping a 3D surface onto a 2D plane while

maintaining injectivity (one-to-one mapping) and low shape distortion. We made **the first** successful attempt to apply a pre-trained model to the fixed-boundary parameterization task and achieved the first feed-forward parameterization. It is worth noting that feed-forward parameterization is a highly challenging task. Even the state-of-the-art neural

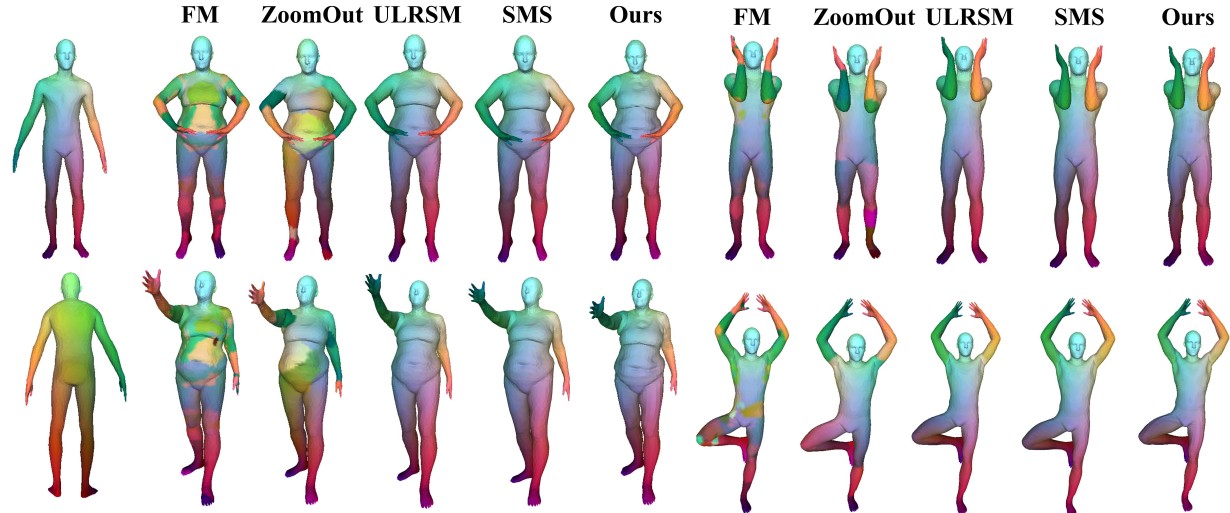

*Figure 10.* Visual comparison of correspondence results by different methods. From left to right: FM (Ovsjanikov et al., 2012), ZoomOut (Melzi et al., 2019), ULRSM (Cao et al., 2023), SMS (Cao et al., 2024), and Ours.

*Table 5.* Quantitative comparison of non-rigid shape correspondence on FAUST. The best results are highlighted in **bold**.

| Method | Input | Error | Time(s) |
|---|---|---|---|
| FM (Ovsjanikov et al., 2012) | Mesh | 21.2 | 3.2 |
| ZoomOut (Melzi et al., 2019) | Mesh | 6.3 | 7.4 |
| ULRSM (Cao et al., 2023) | Mesh | 1.6 | 1.7 |
| SMS (Cao et al., 2024) | Mesh | **1.4** | **0.5** |
| **Ours** | Points | **1.4** | **0.5** |

method FlexPara operates only in an overfitting framework and struggles to achieve global feed-forward parameterization. We experimented with a wide range of feed-forward neural network architectures as well as methods based on various pre-trained models, but none of them could effectively work on the parameterization task.

We made comparisons with the representative traditional geometry optimization method, BFF (Sawhney & Crane, 2017), and the representative neural network-based method, FlexPara (Zhao et al., 2025). Visual comparisons of parameterization results are shown in Figure 13. Our method, together with BFF and FlexPara, produces convincing parameterization outcomes. For quantitative evaluation, we use isometric loss to compare the results. The quantitative comparisons are presented in Table 4. Our method achieves an isometric loss better than BFF and slightly worse than FlexPara. However, thanks to the feed-forward mode, our method achieves considerably faster runtime. Moreover, traditional methods such as BFF require mesh input along with cut seams, whereas our method operates solely on unstructured point clouds.

**Non-Rigid Shape Correspondence.** We experimented with non-rigid 3D shape correspondence on the FAUST (Bogo et al., 2014) dataset. We made comparisons with traditional methods Functional Map (FM) (Ovsjanikov et al., 2012) and ZoomOut (Melzi et al., 2019), as well as learning-based methods ULRSM (Cao et al., 2023) and SMS (Cao et al., 2024). We used the first 80 pairs of the FAUST dataset for training and the last 20 pairs for testing. We reported the average geodesic loss and runtime in Table 5. Our method, using only point cloud as input, achieves the best geodesic distance error while achieving substantially faster runtime, since our approach does not need to compute the Laplace-Beltrami operator. Visualization results are shown in Figure 10.

**Shape Classification.** We experimented with real-scanned 3D object classification on the ScanObjectNN (Uy et al., 2019) benchmark dataset. We made comparisons with existing advanced 3D geometric pre-training frameworks, including OcCo (Wang et al., 2021), CrossPoint (Afham et al., 2022), Point-BERT (Yu et al., 2022), Point-MAE (Pang et al., 2022), Point-M2AE (Zhang et al., 2022), Point-Dif (Zheng et al., 2024), Point-DAE (Zhang et al., 2025b) and PointSD (Chen et al., 2025). To highlight the quality of pre-trained features, we chose to freeze the pre-trained backbone and train the classification head. As shown in Table 6, in the OBJ-BG and OBJ-ONLY settings, our method achieves the near best overall performance. In the PD-T50-RS setting, which includes random rotation data augmentation, our method also reaches the near best performance, indicating that our approach can more effectively capture intrinsic geometric features. To further verify rotation robustness, we set up the PB-T50-RS-Test-Only setting: using

*Table 6.* Quantitative comparison of classification on ScanObjectNN. The best results are highlighted in **bold**. Note that OcCo and CrossPoint do not provide the results on **OBJ-ONLY** and **PB-T50-RS**.

| Method | OBJ-BG | OBJ-ONLY | PB-T50-RS | PB-T50-RS-Test-Only |
|---|---|---|---|---|
| OcCo (Wang et al., 2021) | 84.5 | - | - | 54.1 |
| CrossPoint (Afham et al., 2022) | 86.2 | - | - | 55.7 |
| Point-BERT (Yu et al., 2022) | 88.1 | 87.4 | 83.1 | 53.7 |
| Point-MAE (Pang et al., 2022) | 90.0 | 88.3 | 85.2 | 55.2 |
| Point-M2AE (Zhang et al., 2022) | 91.2 | 88.8 | 86.5 | 58.1 |
| PointDif (Zheng et al., 2024) | 93.3 | 91.9 | 87.6 | 64.2 |
| Point-DAE (Zhang et al., 2025b) | 93.9 | 93.1 | 88.7 | - |
| PointSD (Chen et al., 2025) | **95.2** | **93.6** | **90.1** | - |
| **PTv3 from Scratch** | 89.9 | 88.2 | 84.9 | - |
| **Ours+Point-MAE** | 93.9 | 92.7 | 89.9 | - |
| **Ours** | 93.5 | 92.1 | 89.7 | **72.1** |

*Table 7.* Quantitative comparison of part segmentation on ShapeNetPart. The best results are highlighted in **bold**. Note that OcCo and CrossPoint do not provide standalone pretrained weights that are decoupled from downstream task heads, and PointDif does not include part segmentation fine-tuning results on ShapeNetPart in its original evaluation.

| Method | mIou Freeze | mIou Finetune |
|---|---|---|
| OcCo (Wang et al., 2021) | - | 85.2 |
| CrossPoint (Afham et al., 2022) | - | 85.5 |
| Point-Bert (Yu et al., 2022) | 83.9 | 85.6 |
| Point-MAE (Pang et al., 2022) | 84.3 | 86.1 |
| Point-M2AE (Zhang et al., 2022) | 84.5 | **86.5** |
| PointDif (Zheng et al., 2024) | 84.5 | - |
| Point-DAE (Zhang et al., 2025b) | - | 86.4 |
| PointSD (Chen et al., 2025) | - | 86.1 |
| **Ours** | **85.1** | **86.5** |

weights trained on OBJ-BG, and then testing directly on the PB-T50-RS data which includes rotation augmentation. The results demonstrate that our method exhibits significantly better rotation robustness. additionally, we combined the features from Point-MAE with those from our pre-trained model for a classification task, resulting in observable performance gains.

**Part Segmentation.** We experimented with 3D object part segmentation on the ShapeNetPart (Yi et al., 2016) benchmark dataset under two evaluation protocols: full fine-tuning and training with backbone frozen. The results are shown in Table 7. Under full fine-tuning, our method achieves state-of-the-art performance compared to other methods. Under the frozen backbone setting, our method significantly outperforms other approaches, demonstrating that our pre-trained model provides higher-quality features.

## 5. Conclusion and Discussion

We proposed a novel 3D pre-training paradigm that prioritizes intrinsic geometric properties over extrinsic spatial structures. By defining the pretext task as Geodesic Distance Prediction and enforcing a Geodesic Structure Consistency loss, we isometrically embedded the Riemannian manifold structure into the high-dimensional feature space. We conducted comprehensive experiments to demonstrate that our approach not only achieves high-precision geodesic distance prediction but also shows superior performance on downstream tasks, including fixed-boundary surface parameterization, non-rigid shape correspondence, 3D shape classification and part segmentation.

Our in-depth explorations indicate a highly promising direction of intrinsic geometry learning. Although our method has achieved some progress in rotation robustness, it is still far from sufficient. In the future, we will explore more advanced intrinsic representation learning and attempt to investigate multi-task-driven 3D representation pre-training, further pushing the boundaries of geodesic-guided 3D representation learning.

## Acknowledgment

This work was supported in part by the NSFC under Grant 62422118, and in part by the Hong Kong RGC under Grants 11219324 and N_CityU1114/25.

## Impact statement

This paper presents work whose goal is to advance the field of machine learning. There are many potential societal consequences of our work, none of which we feel must be specifically highlighted here.

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

# A. Model Structure

We employ PTv3 as the backbone and design three scaled versions (Small, Base, and Large) to investigate the scaling behavior of the proposed approach. The detailed configurations are summarized in Table 8. Following the PTv3 backbone, we attach a lightweight MLP regression head to predict a single positive scalar distance. The head takes the extracted feature vector as input and passes it through three successive linear layers. The first two layers are followed by ReLU activations and dropout regularization, while the final linear layer is succeeded by a Softplus activation.

*Table 8.* Architectural configurations of the Small, Base, and Large backbone variants.

| Component | Small | Base | Large |
|---|---|---|---|
| Encoder | `depths` = (2,2,2,6,2) 
 `channels` = (32,64,128,256) 
 `num_heads` = (2,4,8,16) 
 `patch_size` = 1024 (all) | (2,2,2,6,2) 
 (32,64,128,256,512) 
 (2,4,8,16,32) 
 1024 (all) | (3,3,3,12,3) 
 (64,128,256,512,768) 
 (4,8,16,32,48) 
 1024 (all) |
| Decoder | `depths` = (2,2,2,2) 
 `channels` = (256,256,256) 
 `num_heads` = (2,4,8) 
 `patch_size` = 1024 (all) | (2,2,2,2) 
 (512,512,512,512) 
 (2,4,8,16) 
 1024 (all) | (3,3,3,3) 
 (768,768,768,768) 
 (4,8,16,32) 
 1024 (all) |

# B. Implementation of Downstream Tasks

## B.1. Fixed-Boundary Surface Parameterization

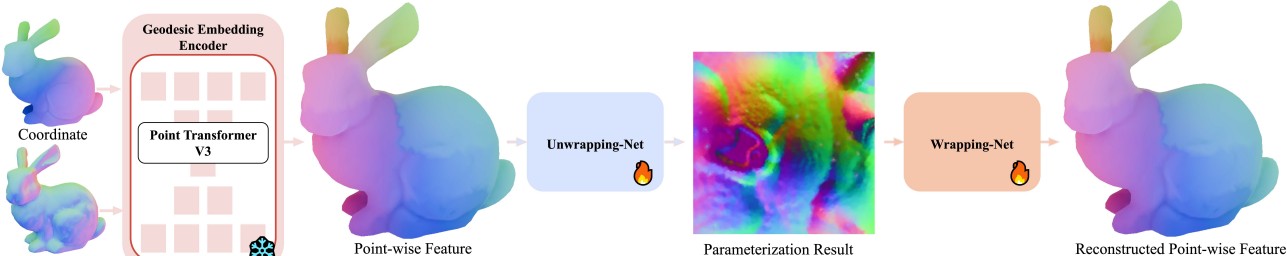

*Figure 11.* The overall pipeline of our fixed-boundary surface parameterization framework.

In the fixed-boundary surface parameterization task, we adopt the unwrapping-wrapping architecture from FlexPara. The raw point cloud is first fed into our pre-trained model to obtain per-point features. These point-wise features are then passed through a lightweight unwrapping module to produce per-point UV coordinates. Subsequently, the UV coordinates are fed into a wrapping module to reconstruct high-dimensional per-point features. Both the unwrapping and wrapping modules are implemented as lightweight MLPs. The framework is shown in Figure 11.

Regarding the loss functions, following FlexPara, we employ a consistency loss to constrain the reconstruction quality after wrapping, and an isometric loss to regularize the deformation in the UV space. Additionally, since this is a fixed-boundary task, we introduce an extra Chamfer Distance (CD) loss $\mathcal{L}_w$ between the predicted UV shape and a regular grid to enforce boundary shape conformity.

$$\mathcal{L}_c = \|\mathbf{F} - \mathbf{F}_{\text{recon}}\|_1 \tag{10}$$

$$\ell_{\text{iso}} = \sum_{\Omega_{\mathbf{P}} \in \mathcal{X}} \sum_i \|\theta_i - \beta_i\|_1 . \tag{11}$$

$$\ell_{\text{global}} = \alpha_w \cdot \ell_w + \alpha_c \cdot \ell_c + \alpha_{\text{iso}} \cdot \ell_{\text{iso}}, \tag{12}$$

As a baseline method, we extend the global parameterization of FlexPara by incorporating the same UV-shape constraint via the CD loss between UV and the grid, while keeping all other settings and model architecture identical to FlexPara. The BFF approach naturally produces rectangular parameterization boundaries. Since it requires an input mesh along with corresponding cut seams, we utilize the pre-provided cut seams from the dataset in *Progressive Parameterizations* (Liu et al., 2018).

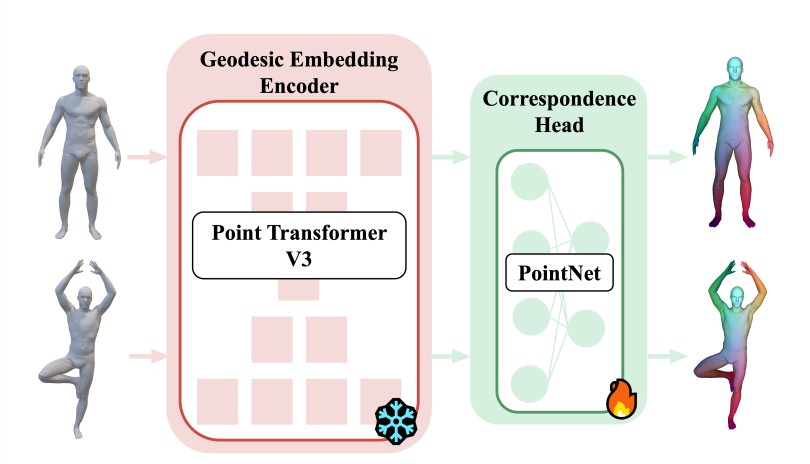

*Figure 12.* The overall pipeline of our 3D shape correspondence framework.

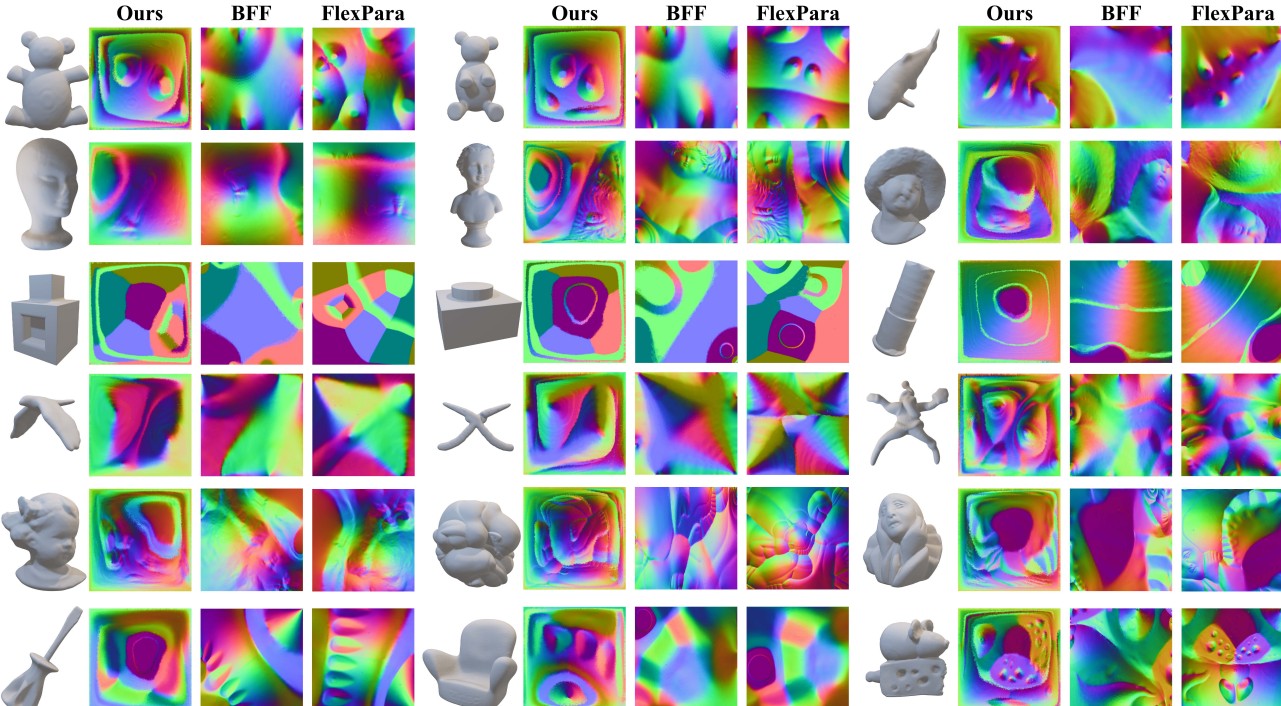

*Figure 13.* More visualization of fixed-boundary surface parameterization results produced by different approaches. From left to right: Ours, BFF and FlexPara.

## B.2. 3D Shape Correspondence

In the shape correspondence task, we utilize a simple PointNet as the decoder head. Experiments were conducted on the FAUST dataset, using the first 80 objects for training and the remaining 20 for testing. The framework is shown in Figure 12. The network takes a 3D point cloud as input and outputs per-point features. A similarity matrix C is computed between the per-point features of two shapes. The model is trained to minimize the discrepancy between C and the identity matrix I.

## B.3. 3D Shape Classification

We evaluate our approach on the ScanObjectNN dataset for the 3D object classification task. In our method, we employ a simple PointNet-style classification head on top of the pre-trained features.

To more fairly demonstrate the quality of features provided by our pre-trained model and to exclude interference from

*Table 9.* Quantitative comparison of fixed-boundary surface parameterization. The best results are highlighted in **bold**.

| Model | BFF | FlexPara | Ours |
|---|---|---|---|
| Input | Mesh + Seams | Points | Points |
| Bear1 | 15.16 | 7.20 | **5.33** |
| Bear2 | 17.15 | 6.69 | **5.13** |
| Fish | 18.31 | **4.52** | 6.15 |
| Head1 | 7.72 | 5.29 | **4.31** |
| Head2 | 11.34 | **5.07** | 6.71 |
| Head3 | 10.23 | 4.20 | **4.18** |
| Box | 24.38 | 4.97 | **4.15** |
| Box2 | 18.76 | 5.99 | **3.78** |
| Nail | 7.43 | 6.23 | **4.11** |
| Bird | 13.48 | 6.41 | **5.62** |
| Tongs | 21.74 | **4.86** | 8.11 |
| Starfish | 16.41 | **5.41** | 8.29 |
| Girl | 8.90 | 6.69 | **3.80** |
| Curl | 17.01 | 7.05 | **3.53** |
| Pensive | 12.08 | 5.95 | **3.75** |
| Screwdriver | 15.48 | **5.71** | 5.75 |
| Sofa | 5.63 | 6.19 | **5.29** |
| Mouse | 20.71 | 4.34 | **3.95** |

PTv3's inherently more advanced modeling capacity, we freeze the pre-trained PTv3 backbone during training and only optimize the lightweight PointNet classification head. In contrast, for all baseline methods, we follow their original settings and perform full fine-tuning of the entire model.

### B.4. 3D Part Segmentation

We evaluate the part segmentation task on the ShapeNetPart dataset. For our method, we adopt a simple MLP as the segmentation head built upon the pre-trained features. In the comparative experiments, we conduct evaluations under two distinct settings: full fine-tuning and frozen backbone. Full fine-tuning represents the common practice for applying pre-trained models to part segmentation tasks. To more rigorously compare the feature quality provided by different pre-training methods and to minimize interference from variations in network architecture, we additionally freeze the pre-trained backbone for each method and train only the segmentation head. This setup isolates the contribution of the learned features to the downstream segmentation performance. Note that OcCo and CrossPoint do not provide standalone pre-trained weights that are decoupled from downstream task heads; therefore, results under the frozen-backbone setting are not available for these methods. Similarly, PointDif does not include part segmentation fine-tuning results on ShapeNetPart in its original evaluation, so no full fine-tuning results are reported for it.

## C. Additional Experimental Results

### C.1. Fixed-Boundary Surface Parameterization

More visual comparison results for fixed-boundary parameterization are presented in Figure 13, with the corresponding isometric quantitative results given in Table 9. It can be observed that, across different types of 3D shapes, our method consistently achieves good fixed-boundary parameterization results, comparable to methods like FlexPara that require long per-scene overfitting times.

### C.2. 3D Shape Correspondence

More visual comparison results for 3D shape correspondence are presented in Figure 14. It can be observed that our method consistently delivers accurate shape correspondence results across different human bodies and a wide variety of poses, achieving performance on par with state-of-the-art task-specific approaches.

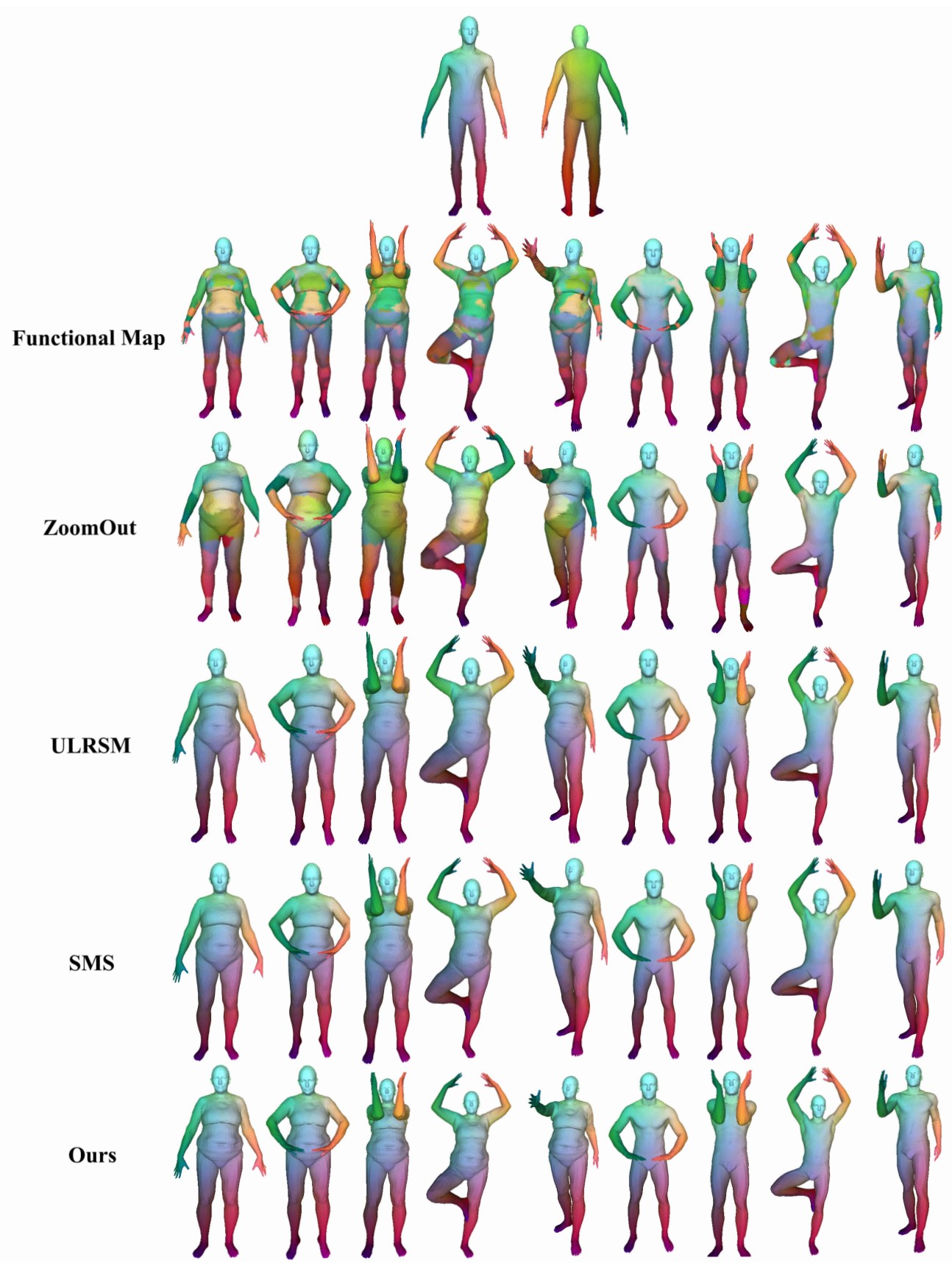

*Figure 14.* More visualization of non-rigid 3D shape correspondence results produced by different approaches. From top to down: FM, ZoomOut, ULRSM, SMS, and Ours.

