# OpenReview forum: "From Extrinsic to Intrinsic: Geodesic-Guided Representation Learning for 3D Geometric Data"
_ICML.cc/2026/Conference — ICML 2026 regular_

### Official Review · Reviewer_Beor · 2026-02-20

**Soundness:** 2
**Presentation:** 3
**Significance:** 3
**Originality:** 3
**Overall Recommendation:** 4
**Confidence:** 3

**Summary:**

This paper introduces PRISM, a self-supervised framework that trains a Point Transformer V3 backbone to predict geodesic distances directly from raw 3D point clouds. Grounded in the Nash Embedding Theorem, it learns an isometric embedding of the underlying manifold through a novel two-stage training process: a structural warm-up followed by importance-sampled fine-tuning to handle naturally imbalanced distance data. The resulting features performs welll on standard semantic tasks and unlock rare feed-forward capabilities for tough geometric challenges like surface parameterization and shape correspondence.

**Compliance With Llm Reviewing Policy:**

Affirmed.

**Final Justification:**

The comprehensive rebuttal has addressed my concerns.  If the paper gets accepted, it is critical that the concept of "isometric embedding" is framed carefully in the final manuscript to ensure precision in terminology.

**Key Questions For Authors:**

1. The geodesic metric may assume a continuous manifold. How does PRISM handle real-world sensor data that might include disconnected components, large holes due to occlusion, or severe topological noise where the true geodesic distance can be mathematically undefined?


2. The importance sampling phase uses a hand-craft weights Eq.(9). Because the probability density drops significantly at extreme distances such as near-zero proximity or the shape’s maximum diameter, this could lead to massive loss weights. Were any specific clipping or normalization strategies used to prevent gradient explosion during fine-tuning?


3. While the paper acknowledges LiteGE (2026) as a recent alternative, a direct comparison is missing. Could you provide a side-by-side analysis of PRISM’s inference time, computational complexity, and MRE/L1 accuracy against LiteGE, especially since their model looks more lightweight?


4. Regarding the 89.7% accuracy on ScanObjectNN (PB-T50-RS), how much of this performance stems from the PRISM pre-training rather than the baseline representational power of Point Transformer V3? This is particularly relevant given that concurrent architectures like DeepLA-Net have reached 90.6% on the same benchmark.


5. The manuscript relies on the Nash Embedding Theorem to justify its "isometric embedding" claims. However, by passing feature differences through a non-linear MLP to estimate distance, the model doesn't strictly guarantee metric space properties like the triangle inequality within the latent space itself. Would it be more accurate to call this a "learned parameterized metric" ?

**Limitations:**

Yes, but the potential scaling limits of the ML decoder when applied to highly complex, multi-object environments (as opposed to isolated, normalized CAD models) may be further discussed.

**Strengths And Weaknesses:**

### Strengths

* Contemporary neural frameworks like FlexPara or traditional algorithms are usually limited in per-scene optimization or require strict meshes with pre-cut seams, but this model hits a 0.5-second inference time using raw, unstructured point clouds. This is a massive jump in efficiency for 3D asset processing.


* There is a clear grasp of domain physics in the training setup. The authors identify that geodesic distances on a 3D manifold follow a normal-like distribution, which causes standard models to underfit critical short- and long-range topological extrema. Addressing this through an importance sampling phase using an empirical probability density function shows a thoughtful approach to manifold geometry.


* Modernizing the pipeline by replacing slow Dijkstra-based wavefront propagation with the scalable Point Transformer V3 architecture makes the pretext task of geodesic distance prediction viable for large-scale learning environments.


### Weaknesses

* The empirical evaluation feels somewhat isolated from the most recent literature, which potentially overstates the model's performance. For instance, the geodesic distance prediction lacks a direct comparison in Table 1 against LiteGE (AAAI 2026), a method the authors acknowledge in their related work as being highly efficient for sparse point clouds. Similarly, while the 89.7% accuracy on ScanObjectNN is competitive within the paper's provided context, it trails behind some recent SOTA architectures like DeepLA-Net (90.6% in their paper).


* The branding of the latent space as an "isometric embedding" is theoretically loose. Although the authors lean on the Nash Embedding Theorem for justification , the actual implementation seems to directly regress scalar distances by passing feature differences through a non-linear MLP. This setup maps the data into a learned pseudo-metric space rather than a strict one. Since the non-linear MLP doesn't mathematically guarantee standard metric properties like the triangle inequality within the features themselves, the claim of true isometry could be technically inaccurate.

---

> ### Author Rebuttal · Authors · 2026-03-31
>
> Pictures and tables are in https://anonymous.4open.science/r/ICML26-rebuttal-B8D8 .
>
> **W1\&Q3:**
> We emphasize that our PRISM outperforms LiteGE in geodesic distance prediction in Table A. We conducted comparisons on the same test dataset used by LiteGE. **Note that LiteGE's computation of the L1-based error metric is different from ours.** Hence, Table 1 includes both the standard MRE metric and LiteGE's L1-based error metric, where our approach consistently shows significant advantages.
>
> In our comparative experiments on downstream tasks, we primarily focused on evaluating the performance of mainstream pre-training models. The experimental results demonstrate that our method is highly competitive. DeepLA-Net is a more powerful network architecture rather than a pre-training method; therefore, we did not include it in this specific comparison.
>
> **W2 \& Q5:**
> You may have **overlooked** the geodesic structure loss in Eq. (6), which is used
> to constrain the structural similarity between the feature space and the geodesic distance. In Figure C, we show a heatmap of geodesic distance and feature Euclidean distance between 50 points on Bunny. It can be seen that the distribution and scale ordinary between geodesic and feature are similar. In Figure A, we also present a comparative analysis of training dynamics. By comparing our method with a variant that calculates feature-wise Euclidean distance without an MLP, it becomes evident that the latter suffers from convergence issues, whereas our full model converges smoothly.
>
> We acknowledge your point and agree that referring to the relaxed training paradigm strictly as "isometric embedding" may not be entirely appropriate. Isometric mapping serves as our training objective rather than a hard constraint strictly enforced within the network. Replacing it with a more accurate terminology is a very reasonable suggestion, and we will carefully revise this wording in the subsequent version of the manuscript. We sincerely appreciate your constructive feedback.
>
> **Q1:** We present a set of results featuring non-watertight and noisy inputs. As
> illustrated, our method demonstrates strong robustness against both topological
> defects and sensor noise.
>
> **Q2:** Yes, we employ gradient clipping to prevent gradient explosion during training.
>
> **Q4:** Regarding the source of our performance improvements, we train a PointTransformer V3 on ScanObjectNN from scratch. As can be seen, PRSIM gives a better result than PTv3 from scratch. This demonstrates that the PRISM training method itself is highly valuable. Naturally, employing a more powerful backbone—such as DeepLA-Net—would bring further performance gains, but architecture engineering is not the primary focus of this paper.

---

> > ### Author Rebuttal · Reviewer_Beor · 2026-04-02
> >
> > I sincerely thank the authors for their comprehensive rebuttal, which has addressed my concerns.  If the paper gets accepted, it is critical that the concept of "isometric embedding" is framed carefully in the final manuscript to ensure precision in terminology.

---

> > > ### Author Response · Authors · 2026-04-03
> > >
> > > We are glad to have cleared up your concerns and thank you for the positive assessment. We will carefully review and improve the concept you pointed out in the subsequent version of the manuscript.

---

### Official Review · Reviewer_9qFn · 2026-03-06

**Soundness:** 3
**Presentation:** 3
**Significance:** 3
**Originality:** 3
**Overall Recommendation:** 4
**Confidence:** 3

**Summary:**

This paper presents the PRISM framework, aiming to achieve self-supervised pre-training of 3D point clouds by Recovering the Intrinsic Surface geodesic Metric. The core logic is to map the manifold structure to the feature space by using the Nash embedding theorem through Geodesic Regression Loss, Mean Relative Error Loss and Geodesic Structure Consistency Loss. The paper claims that it has significant advantages in handling non-rigid deformation and feedforward parameterization tasks.

**Compliance With Llm Reviewing Policy:**

Affirmed.

**Final Justification:**

I recommend Weak Accept (4). The authors have addressed the initial weaknesses regarding non-equidistant deformation and architectural constraints. The implementation of the first feed-forward surface parameterization model is a high-impact contribution that solves a long-standing bottleneck in geometry processing. Given the strong empirical results across classification, segmentation, and correspondence tasks, this paper is technically solid and highly significant.

**Key Questions For Authors:**

Questions:

1. In Formula (6), the scaling factor $\alpha$, as the steepness control parameter of the soft-sign function, what specific impact does it have on gradient convergence? How was it determined in the experiment?

2. What is the dynamic allocation logic in the two-stage training? In particular, how is the turning point for disabling structural loss to enter the fine-tuning stage defined?

3. Real scan data often contain noise and non-manifold structures. The MMP algorithm usually requires a watertight manifold. How sensitive is this method to the topological quality of the input point cloud? If there are holes or noise in the input data, will the robustness of geodesic distance prediction significantly decline?

4. The paper emphasizes learning the intrinsic geometric properties. In actual feature visualization, how can it be proved that the learned features indeed decouple the internal geometry from the external posture? Apart from t-SNE visualization, are there more quantitative indicators to evaluate the degree of isometric maintenance in the feature space?

**Limitations:**

yes

**Strengths And Weaknesses:**

Strengths:

1. Compared with methods that require scene-by-scene optimization (such as NeuroGF or FlexPara), this method has a magnitude advantage in inference speed (0.5 seconds versus hundreds of seconds).

2. The experiments cover a wide range of tasks from high-level semantics (classification/segmentation) to low-level geometry (parameterization/correspondence).

3. The first Feed-forward surface parameterization model has been successfully implemented. This has changed the previous situation where this task had to rely on time-consuming per-scene optimization.

Weaknesses:

1. The model uses the absolute value of the difference between feature vectors to ensure the symmetry of distance prediction. Although this approach conforms to the metric standards, it may overlook the directional characteristics on the manifold and may not be precise enough for the description of some asymmetric local geometries.

2. The paper mainly emphasizes the invariance under equidistant deformation. However, in practical applications, non-rigid objects often undergo non-equidistant deformations (such as stretching and contracting), and the performance of the model in dealing with such significant tensile deformations has not been fully verified.

---

> ### Author Rebuttal · Authors · 2026-03-31
>
> Pictures and tables are in https://anonymous.4open.science/r/ICML26-rebuttal-B8D8 .
>
> **W1:**
> The "symmetric predictor" we mentioned refers to ensuring that the network's output is independent of the input order of the two query points, rather than referring to geometric symmetry. More importantly, it should be clarified that the operation of deducing absolute feature difference is just part of our geodesic distance prediction head atop the PTv3 encoder features. The feature extraction mechanisms and geometric representation capabilities of the standard PTv3 backbone are not destroyed. Thus, our approach naturally leverages PTv3's expressiveness in handling complex shapes, and is by no means restricted by an inability to represent asymmetric geometries.
>
> **W2:** Regarding the issue of non-isometric shapes, our learned representation features maintain robust perceptual capabilities, provided that the deformation (e.g., stretching) of the object is not so severe as to fundamentally disrupt the geodesic distance structure. This is evidenced by the shape correspondence experimental results detailed in the Appendix, which demonstrate that our extracted features remain highly effective for establishing correspondence across human bodies of varying builds and proportions.
>
> **Q1:** According to our empirical observations, an excessively large $\alpha$ can lead to instability during the initial stages of training. To determine the optimal value, we conducted experiments with several different $\alpha$ settings while keeping other hyperparameters and datasets constant. Our results indicate that setting $\alpha=10$ maintains training stability and effectively accelerates convergence.
>
> **Q2:** The primary objective of importance sampling fine-tuning is to address the inaccurate geodesic distance estimation for both near and far query points. We observed that introducing importance sampling too early in the training process often results in convergence difficulties. Therefore, we opt to initiate importance sampling as a fine-tuning stage only after the model has fully converged without it.
>
> **Q3:** In Figure B, we present a set of results featuring non-watertight and noisy inputs. As illustrated, our method demonstrates strong robustness against both topological defects  and sensor noise.
>
> **Q4:** In the classification task, we have demonstrated the rotational robustness of our approach in Table 6 of the manuscript. Compared to other baseline methods, our model exhibits significantly stronger robustness against rotations. In Figure C, we show a heatmap of geodesic distance and feature Euclidean distance between 50 points on Bunny. It can be seen that the distribution and scale ordinary between geodesics and features are similar.

---

> > ### Author Rebuttal · Reviewer_9qFn · 2026-04-02
> >
> > Thank you for your reply, which has solved my problem. I will keep my score.

---

> > > ### Author Response · Authors · 2026-04-02
> > >
> > > We are glad that the previous concerns were resolved to your satisfaction. Thank you for your positive assessment and support.

---

### Official Review · Reviewer_GYGT · 2026-03-10

**Soundness:** 2
**Presentation:** 3
**Significance:** 3
**Originality:** 3
**Overall Recommendation:** 4
**Confidence:** 3

**Summary:**

The paper proposes a self-supervised pretraining method that computes an embedding of 3D shapes according to their geodesic distance in a high-dimensional latent space (for a sufficiently high dimension, this embedding must exist). Technically, it trains a 3D point cloud transformer model to yield embedding vectors that match geodesic distance with good relative L1 error (with some extra tricks such as ordering constraints, uniform sampling over different distance brackets, and modified weighting in early training to speed-up convergence). The concrete method is plausible and not very involved, and all of the major additions are validated by ablation studies.

As a result, the model can predict geodesic distances of shapes using the embedding computed by the feed-forward transformer model, and it beats earlier learning-based work. Advantages over traditional geometry processing techniques can be seen in terms of wall-clock time at medium accuracy (not up to numerical precision, obviously).

Most impressive is the set of downstream tasks demonstrated: The learned representation allows for isometry-invariant correspondence estimation and, as far as the paper claims for the first time, a simple generalizable, feed-forward surface parametrization network. The authors also report favorable results for refining the representation on 3D segmentation and classification tasks.

**Compliance With Llm Reviewing Policy:**

Affirmed.

**Final Justification:**

After reading the rebuttal and the other reviews, I will keep my score of 4 - weak accept. The paper makes some very interesting observation on pretraining geometry, which makes me believe it is worth publishing. It also shows for the first time generalizing feed-forward parametrization, and this result seems very plausible as such to me. However, the more far-reaching results on classification/segmentation are quite surprising and I would feel more comfortable if the background and causes/reasons had been examined more in depth; thus, my positive recommendation is not yet very strong, although the potential for even more interesting novel insights seems quite high.

**Key Questions For Authors:**

My main concern is the lack of details for the downstream tasks (training, evaluation, comparison to previous work). For example, could the refinement step itself already do "all the heavy lifting"? How "usable" are the surface parametrization results (even if there are shortcomings, the fact that it does work to some degree is nice)?

Specifically for the classification (and segmentation) results, I would like to hear a mechanistic interpretation: Why would or should this work (why would pretraining on geodesic distances help here)? Even if this does help to the degree reported (which I do not doubt, I just want to understand it), what is the main effect at play? Could it be that something completely different affects the results (for example, an implicit resampling of the shape according to features, which has been shown to successfully improve results in prior work, see for example methods that pre-cluster the "most interesting" local features on shapes to boost performance and then just run a simple point-net to get SOTA-performance (a reference that comes to my mind is LocalNet by Bytyqi et al, arXiv:2006.07226, but there are previous/more prominent papers that observe the same; I am just bringing this up to put out the hypothesis that it is not possible that it is the geodesic structure that is decisive here).

**Limitations:**

As I already discussed above, I am missing a critical assessment of potential causes for the observed downstream performance. Understanding better why the method works would be very helpful. Similarly, it would be useful to explicitly demonstrate where it breaks (for example for synthetic out-of-distribution shapes).

In terms of social impact, I see no lack of discussion of significant acute or foreseeable aspects.

**Strengths And Weaknesses:**

When first reading the paper, the method does not seem to be very exciting or original - the paper basically trains a powerful transformer model on preserving geodesic distances in feature space, with some engineering applied to task-appropriate weighting. While the mechanics of the pretraining appear unspectacular, the broad success on downstream tasks is indeed quite impressive. If this holds comprehensively, I would think that this is a strong paper. I would thus discuss this aspect in more detail (as the the actual training pipeline seems uncontroversial, but also not very surprising):

- Approximating geodesics: This is an obvious application; to fully understand the advantage over previous methods, it might be useful to also specify the architectural and computational effort (how many flops are used). This is also a point to keep in mind when comparing to traditional methods such as Dijkstra variants or heat flow - back then, the algorithms did not have 8 H100 GPUs at their disposal. Overall, this aspect looks solid, but not very exciting.

- Surface parametrization: This application is structurally related to the pretraining task and thus plausible, but nonetheless it seems to be a very good idea and a major strength of the submission. Unfortunately, the paper is sparse on architecture, training protocol and evaluation. The appendix provides more information, but it would be very useful to discuss parametrization quality and some key aspects of the training already in the main paper. Even in the appendix, the discussion of the quality of the results is brief (are the maps injective? what distortion measures are obtained over the training ground-truth? in how far does it generalize beyond the training data?)  Assuming that this is really the first method of this kind (I am not actively following this area), this would nonetheless be a strong argument in favor of the submission.

- Shape correspondences: This application is again more canonical; I found the evaluation convincing. The description of the technical details is limited to a short section in the appendix; it would be very helpful to add more background information here.

- Classification and segmentation: At this point, I am actually very surprised; it is not immediately technically plausible to me why an isometry-preserving embedding should be useful to enhance shape classification (or segmentation as a closely related task); shapes of similar functionality can have quite different intrinsic geometry (such as biplanes and jets in ModelNet. For ScanObjNN (measured point clouds), I would further expect issues arising from topological noise (due to heavy noise, outliers, often very incomplete acquisition). I am hesitant to believe that the proposed method yields SOTA results by pretraining on an unrelated objective (and then just plugging a point-net on top for refinement, according to the appendix). Here, I would at least like to see some more exploration into why this would or should work.

Generally speaking, the introduction presents the intrinsic view of the geometry to some degree as a superior representation; I am not convinced that that is the case and would suggest a more cautious wording; intrinsic structures can also be fragile (for example, from topological noise, due to modeling or -notoriously- bad acquisition of real-world data), and it emphasizes one specific aspect of a shape (which is useful when matching walking characters, but less so for relating strongly varying geometry; also, for some features, such as small cavities and bends on or in shapes, intrinsic representations might be insensitive, and large scale features might be deformed heavily on a global scale due to the focus on the metric structure). The paper seems to suggest that geodesic pretraining is a catch-all method for various geometry processing tasks, which is a very strong claim. This could indeed be the case and that would be a strong and surprising result, but, because of the potential impact, I would like to see a deeper examination of the tasks that are not related to intrinsic geometry (classification, segmentation), or otherwise I would suggest to position the results as - to some degree - preliminary.

Overall, the paper is quite exciting; so I would see a positive overall impression at this point. The main reason for this are the non-obvious downstream applications such as, in particular, surface parametrization. My impression would be better if these were explored and evaluated (and in the best case, to some extend mechanistically/causally explained) in more detail.

---

> ### Author Rebuttal · Authors · 2026-03-31
>
> Pictures and tables are in https://anonymous.4open.science/r/ICML26-rebuttal-B8D8 .
>
> **1. Model Architecture:** The model structure has been shown in the Appendix. The FLOPs and paramerters are shown in Table D.
>
> **2. Parameterization and Shape Correspondence:**
> Regarding parameterization, to the best of our knowledge, our work is the first to successfully achieve feed-forward parameterization based on pre-trained models. In terms of injectivity, we indirectly constrain the mapping through a wrap-unwrap architecture. If the UV flattening fails to satisfy injectivity, the reconstruction loss of the unwrapped output becomes significantly high—a strategy inspired by established works such as FlexPara and FAM. As for generalization, we partitioned the dataset provided by Progressive Parameterizations. By training on only a small subset, our model generalizes effectively to the entire dataset, including geometrically complex cases like Mouse and Screwdriver. Regarding distortion metrics, we employed the most stringent isometric metric to evaluate UV distortion. The results demonstrate that our method achieves lower distortion across most examples, even under the feed-forward setting.
>
> For shape correspondence, we employ a streamlined PointNet-based architecture. The Global Feature Extraction network comprises layers with dimensions of [768, 512, 256], while the Point-wise Feature Extraction module follows a dimension sequence of [1024, 512, 256, 128], ultimately yielding a 128-dimensional point-wise feature for each point. A similarity matrix C is then constructed by computing the cosine similarity between these features. Finally, we apply an L1 loss constraint between the similarity matrix C and the identity matrix I.
>
> **3. Classification and Segmentation:** We agree that our method is not a universal solution. In certain high-level semantic tasks, an approach solely based on geodesic distance may fall short. However, we would like to emphasize that geodesic-based features can still provide valuable insights for high-level semantic understanding in specific scenarios.
>
> Admittedly, the gap between semantic representations and intrinsic geometric properties can sometimes lead to the failure of our method when capturing semantic features. A prominent example is human pose detection, where our approach is almost entirely ineffective. This is because the geodesic distance structures of the same human body in two different poses can remain basically identical, rendering intrinsic quantities uninformative in this specific scenario. Nevertheless, in our current evaluations—such as the classification and segmentation tasks—we have not observed this type of failure.
>
> Our core contribution lies in the fine-grained capture of intrinsic geometry. The design of our downstream experiments reflects this focus: we first tackle tasks strictly sensitive to fine-grained intrinsic geometry (i.e., parameterization and shape correspondence) to demonstrate our model's capacity to capture these properties effectively. Subsequently, we explore high-level semantic perception tasks (i.e., classification and segmentation) to illustrate that our approach can also offer meaningful improvements in such contexts. In segmentation tasks, semantic parts often align with underlying geometric structures, such as the wings of an airplane. Similarly, in classification tasks, salient geometric features—such as the slender legs of a chair—are often the deciding factors for category identification.
>
> Therefore, as a future best practice, we anticipate that employing a multi-task learning framework could further enhance performance on downstream tasks. To this end, we have provided an additional experiment where we combined the features from Point-MAE with those from our pre-trained model for a classification task in Table C, resulting in observable performance gains. Although preliminary, this experiment highlights the efficacy of our method and demonstrates its complementary value to current mainstream reconstruction-based approaches.

---

> > ### Author Rebuttal · Reviewer_GYGT · 2026-04-03
> >
> > Thanks for the detailed reply. My questions have been resolved within the scope of this paper and what the rebuttal can realistically achieve. I still think that the paper and the discussion in this round of reviews is not yet able to fully explore and explain the success of the method for non-metric downstream tasks such as segmentation and classification, which I personally found to be the most surprising findings. This would probably require extensive experiments and a new paper.
> > I keep would keep my score - which is positive due to the very interesting results; it could be stronger if better/deeper insights into the mechanisms behind the success were obtained, and the potential for artifacts or distortions would have been ruled out more thoroughly. Nonetheless, the findings appear valuable and surprising, so I would see value in publishing the paper for others/future work to explore this further.
> > Please interpret my remaining skepticism not in a negative but in a positive sense - my concern is more of the type "extraordinary findings need extraordinary evidence" rather than lack of interest/significance.

---

> > > ### Author Response · Authors · 2026-04-03
> > >
> > > We appreciate your highly positive evaluation of the significance and value of our work, and we are very pleased with your interest in our paper. As you noted, our work reveals the immense potential of geodesic-driven 3D foundation models, providing the community with insights into a novel and promising paradigm for 3D pre-training.
> > >
> > > As you suggested, following this paper, we plan to launch a new project to evaluate the effectiveness of geodesic-driven 3D foundation models across a wider variety of tasks and domains (not limited to classification, segmentation, correspondence, and parametrization, but also including generation, editing, etc.). This will allow us to expand the applicable domains of geodesic distance as a learning objective and further clarify its effective scope and underlying mechanisms.
> > >
> > > Thank you again for your high appraisal and constructive suggestions regarding our work.

---

### Official Review · Reviewer_ib1A · 2026-03-12

**Soundness:** 3
**Presentation:** 3
**Significance:** 2
**Originality:** 2
**Overall Recommendation:** 4
**Confidence:** 4

**Summary:**

This submission explores a general area of intrinsic geometry learning for 3D representation learning. The manuscript considers a central concept of learning geodesic-aware representations through geodesic distance prediction. Specifically, the paper proposes a pretraining framework named PRISM, which encourages learned point features to preserve intrinsic geometric structure by predicting geodesic distances between pairs of points. The method uses a Point Transformer backbone to extract point-wise features and introduces a training objective composed of geodesic regression loss, relative error loss, and a geodesic structure consistency loss. A two-stage training strategy is further proposed to mitigate the imbalance in geodesic distance distributions. The learned representation is evaluated on several downstream tasks, including surface parameterization, non-rigid shape correspondence, classification, and part segmentation.

**Compliance With Llm Reviewing Policy:**

Affirmed.

**Final Justification:**

I sincerely thank the authors for their detailed rebuttals, which answered my questions. I will increase the score. I hope to revise the relevant rebuttals in the final manuscript.

**Key Questions For Authors:**

1、Does the learned feature space satisfy an approximate isometric constraint (e.g., $\hat{d}_{ij} \approx d_G(p_i,p_j)$)? ? If not, how is intrinsic geometry preserved in the embedding space?

2、How sensitive is the training process to errors in mesh reconstruction or geodesic computation?

3、Could the authors compare their approach with LiteGE, Point-DAE, and Harnessing Text-to-Image Diffusion Models for Point Cloud Self-Supervised Learning. LiteGE also focuses on efficient geodesic embedding?

4、How does the method perform when training without mesh-based geodesic supervision?

**Limitations:**

yes

**Strengths And Weaknesses:**

Strength：The proposed framework attempts to provide a unified pretraining objective that may benefit multiple downstream tasks involving geometric reasoning.

Weaknesses：

1、The paper emphasizes that most existing 3D representation learning methods rely primarily on extrinsic spatial structures and do not explicitly model intrinsic geometry. However, intrinsic geometry has been extensively studied in both classical geometry processing and learning-based shape analysis. In particular, a number of recent works already explore learning representations based on geodesic distances or intrinsic metrics, including methods such as GeGNN, NeuroGF, and the recent LiteGE.

While the proposed framework integrates geodesic prediction into a pretraining pipeline, the manuscript would benefit from a clearer discussion of how PRISM differs conceptually and technically from these existing intrinsic geometry learning approaches. At present, the methodological novelty relative to these prior works is not entirely clear.

2、The paper motivates the approach using the Nash embedding theorem and aims to learn an approximately isometric embedding of the manifold. However, the proposed model predicts geodesic distance through an MLP operating on feature differences: Eq. 3. This formulation allows the MLP to learn an arbitrary mapping between feature space and geodesic distance. As a result, the model may successfully predict geodesic distances without the feature space itself preserving intrinsic geometry. In other words, the intrinsic geometry may not actually be preserved in the learned representation.

3、Geodesic distance is fundamentally defined on a continuous surface manifold, while the proposed framework predicts geodesic distances directly from point clouds. During training, ground-truth geodesic distances are computed from meshes using classical algorithms. However, during inference the model receives only point clouds. A point cloud does not uniquely define the underlying surface geometry, and different surface reconstructions could lead to different geodesic distances. Therefore, the learning problem may not be well-posed without additional assumptions about the underlying surface.

4、The proposed framework performs worse than several recent methods, including LiteGE (2026), Point-DAE (2025), and Harnessing Text-to-Image Diffusion Models for Point Cloud Self-Supervised Learning (2025).

5、The paper demonstrates improvements on semantic tasks such as object classification and part segmentation. Intrinsic features are inherently invariant to isometric transformations (e.g., changes in pose). While this is beneficial for correspondence, it may discard crucial discriminative information for semantic tasks. The authors should clarify the trade-off between geometric invariance and semantic discriminability.

---

> ### Author Rebuttal · Authors · 2026-03-31
>
> Pictures and tables are in https://anonymous.4open.science/r/ICML26-rebuttal-B8D8 .
>
> **W1 \& Q1:** Overall, our main contribution lies in proposing a novel geodesic-distance-based representation learning framework and an improved training paradigm, including importance sampling and geodesic structure loss. Importance sampling is designed to address the imbalance in geodesic distance data, and geodesic structure loss is designed to constrain the structural similarity between the feature space and the geodesic distance.
>
> We show that point-wise features learned under geodesic distance supervision can significantly benefit a wide range of downstream tasks. In contrast, existing works such as GEGNN, LiteGE, and NeuroGF focus more on the geodesic distance prediction task itself. Their limited exploration of downstream applications is mostly confined to directly utilizing the predicted geodesic distances, rather than addressing point cloud feature representation or network fine-tuning.
>
> **W2:** You may have **overlooked** the geodesic structure loss in Eq. (6), which is a critical design for constraining the structural similarity between the feature space and the geodesic distance. In Figure C, we show a heatmap of geodesic distance and feature euclidean distance between 50 points on bunny. It can be seen that the distribution and scale ordinary between geodesic and feature are similar. In Figure A, we also present a comparative analysis of training dynamics. By comparing our method with a variant that calculates feature-wise Euclidean distance without an MLP, it becomes evident that the latter suffers from convergence issues, whereas our full model converges smoothly.
>
> **W3:** We agree that predicting geodesics from point clouds theoretically suffers from multi-solution issues, but it does not mean this task is unsuitable for pre-training. In fact, in the field of representation learning, pre-training tasks (such as masked auto-encoders for image modeling and next token prediction for language modeling) are typically ill-posed, where the optimization targets are not fully deterministic mappings. In our framework, the backbone is driven to learn informative geometric priors, and downstream experiments also demonstrate our effectiveness.
>
> **W4, W5 \& Q3:** We emphasize that our PRISM outperforms LiteGE in geodesic distance prediction in Table A. We conducted comparisons on the same test dataset used by LiteGE. **Note that LiteGE's computation of the L1-based error metric is different from ours.** Hence, Table A includes both the standard MRE metric and LiteGE's L1-based error metric, where our approach consistently shows significant advantages.
>
> As shown in Table B and Table C, our method maintains a leading position in segmentation tasks. In terms of classification, our performance is slightly inferior to PointSD but remains comparable to PointDAE. For high-level tasks, our approach admittedly exhibits certain disadvantages compared to cross-modal and reconstruction-based methods, as our model focuses heavily on fine-grained geometric details. However, in low-level domains, such as parameterization tasks, our method demonstrates substantial advantages and stands as the first successful pre-trained model for this specific application. For future best practices, our method could potentially be integrated with state-of-the-art high-level methods for multi-task pre-training. We conducted an experiment by combining the features from Point-MAE and PRISM for a classification task. The results showed a noticeable performance gain. While this is a preliminary study, it potentially demonstrates the effectiveness of our approach and its complementarity to existing reconstruction-based methods.
>
> **Q2 \& Q4:** The reviewer appears to have misunderstood our workflow of ground-truth data preparation. For training, we directly compute geodesic distances from existing mesh data (which have been pre-processed to be watertight manifolds), instead of reconstructing meshes from point clouds for computing geodesics. Besides, the adopted MMP algorithm, as an exact geodesic computation method, does not introduce approximation errors.
>
> Given the vast availability of mesh data itself and the fact that using traditional point-cloud-based geodesic algorithms to generate ground-truth may lead to obvious errors, we believe that using mesh-based geodesic supervision is a very natural and reasonable choice.

---

> > ### Author Rebuttal · Reviewer_ib1A · 2026-04-03
> >
> > I sincerely thank the authors for their detailed rebuttals, which answered my questions. I will increase the score. I hope to revise the relevant rebuttals in the final manuscript.

---

> > > ### Author Response · Authors · 2026-04-06
> > >
> > > We are pleased to see that your concerns have been addressed. Thank you for your recognition of our work and for increasing the score, we will incorporate the necessary explanations from the rebuttal into the revised manuscript.

---

### Decision · Program_Chairs · 2026-04-30

**Decision:**

Accept (regular)

**Comment:**

This submission received mixed reviews initially (3, 4, 4, 4). After the rebuttal and discussion, the negative reviewer increased their score, resulting in 4x unanimous weak accept (4) scores.

The reviewers agree that the paper presents a valuable pretraining method for learning geodesic-aware 3D representations, which (somewhat surprisingly) prove effective for various downstream applications. Concerns were raised in the reviews regarding the positioning and framing of the proposed method and the lack of clarity in technical details and interpretation of the results.

The rebuttal cleared up most of the questions, and all reviewers find the merits of the paper worth publishing. The Area Chairs agree with this assessment and recommends Accept, but encourages the authors to incorporate the feedback into the revision.